# Stability and Deviation Optimal Risk Bounds with Convergence Rate $O(1/n)$

**Yegor Klochkov**
Cambridge-INET, Faculty of Economics
University of Cambridge
yk376@cam.ac.uk

**Nikita Zhivotovskiy**
Department of Mathematics
ETH, Zürich
nikita.zhivotovskii@math.ethz.ch

## Abstract

The sharpest known high probability generalization bounds for uniformly stable algorithms (Feldman, Vondrák, NeurIPS 2018, COLT, 2019), (Bousquet, Klochkov, Zhivotovskiy, COLT, 2020) contain a generally inevitable sampling error term of order $\Theta(1/\sqrt{n})$. When applied to excess risk bounds, this leads to suboptimal results in several standard stochastic convex optimization problems. We show that if the so-called Bernstein condition is satisfied, the term $\Theta(1/\sqrt{n})$ can be avoided, and high probability excess risk bounds of order up to $O(1/n)$ are possible via uniform stability. Using this result, we show a high probability excess risk bound with the rate $O(\log n/n)$ for strongly convex and Lipschitz losses valid for *any* empirical risk minimization method. This resolves a question of Shalev-Shwartz, Shamir, Srebro, and Sridharan (COLT, 2009). We discuss how $O(\log n/n)$ high probability excess risk bounds are possible for projected gradient descent in the case of strongly convex and Lipschitz losses without the usual smoothness assumption.

## 1   Introduction

Stability is a standard method to analyze the generalization properties of learning algorithms. This approach can be traced back to the foundational works of Vapnik and Chervonenkis [46]. Using the sensitivity of the learning algorithms to the removal of one example in the learning sample, they proved optimal bounds (scaling as $O(1/n)$, where $n$ is the sample size) on the average risk of *hard margin SVM* and of the Perceptron algorithm. The ideas of stability were further developed by Rogers and Wagner [40], Devroye and Wagner [13, 14], Lugosi and Pawlak [33], Kearns and Ron [26] and other authors. Stability arguments are notorious for only providing in-expectation error bounds. High probability guarantees require more effort and lead to several long-standing open problems in the literature. For example, the classical stability analysis of Vapnik and Chervonenkis [46, 22] has only recently been refined to allow high probability guarantees with the optimal error rate [50, 8, 19].

The widely used notion of stability allowing high probability upper bounds is called *uniform stability*. It was introduced in the seminal work of Bousquet and Elisseeff [7]. Let us introduce some standard notation. We have a set of $n$ i.i.d. observations $S = \{X_1, \ldots, X_n\}$ sampled according to some unknown distribution $P$ defined on an abstract set $\mathcal{X}$. One may naturally think of $\mathcal{X}$ as a set of instances with their labels. Our decision rules are indexed by a set $\mathcal{W}$ that is always assumed to be a closed subset of a separable Hilbert space. Given the *learning sample* $S$, a *learning algorithm* produces the decision rule $w_n = w_n(S) \in \mathcal{W}$. For the loss function $\ell : \mathcal{X} \times \mathcal{W} \to [0, \infty)$, we define the risk and the empirical risk of $w \in \mathcal{W}$, respectively as

$$R(w) = \mathbb{E}_P \ell(X, w), \qquad R_n(w) = \frac{1}{n} \sum_{i=1}^{n} \ell(X_i, w).$$

35th Conference on Neural Information Processing Systems (NeurIPS 2021).

Following [7], an algorithm $w_n$ (we always use the word *algorithm* both for the mapping and for the decision rule) is uniformly $\gamma$-stable, if for any $x, x', x_1, \ldots, x_n \in \mathcal{X}$ and $i = 1, \ldots, n$, it holds that

$$|\ell(x, w_n(x_1, \ldots, x_n)) - \ell(x, w_n(x_1, \ldots, x_{i-1}, x', x_{i+1}, \ldots, x_n))| \leq \gamma.$$

The paper of Hardt, Recht, and Singer [20] on the stability of gradient descent methods has generated a wave of interest in this direction. Recent works use various notions of stability in their analysis: some authors are motivated by the analysis of gradient descent algorithms [32, 30, 16, 5], while others use the notion of average stability to obtain the in-expectation $O(1/n)$ rate for regularized regression [29, 18, 47] and some more specific *improper* learning procedures [37, 38]. One of the key open questions left in [20] is related to the lack of high probability generalization bounds. Their question inspired a line of research focused on getting the sharpest possible generalization bounds for uniformly stable algorithms. Based on the recent progress by Feldman and Vondrák [16, 17], the sharpest known high probability bound for uniformly stable algorithms was shown by Bousquet, Klochkov and Zhivotovskiy [9]. Their result states that for a $\gamma$-uniformly stable algorithm $w_n$ if the loss $\ell(\cdot, \cdot)$ is bounded by $M$, then for any $\delta \in (0, 1)$, with probability at least $1 - \delta$, it holds that

$$R(w_n) - R_n(w_n) \lesssim \underbrace{\gamma \log n \log\left(\frac{1}{\delta}\right)}_{\text{stability error}} + \underbrace{M\sqrt{\frac{1}{n} \log\left(\frac{1}{\delta}\right)}}_{\text{sampling error}}. \tag{1}$$

One problem inherent to *all* high probability generalization bounds is that they are insensitive to the stability parameter $\gamma$ being smaller than $M/\sqrt{n}$. That is, in this favorable case the *sampling error* term scaling as $O(1/\sqrt{n})$ controls the generalization error. The situation where $\gamma$ is smaller than $M/\sqrt{n}$ happens in the literature on stochastic convex optimization where the strongly convex objectives are frequently considered [42, 43, 39, 21]. Unfortunately, there is no generic way to remove the $O(1/\sqrt{n})$ term in (1). It appears even for the algorithms that always output the same decision rule (0-uniform stability). The problem is that generalization bounds compare the finite-sample risk $R_n$ with its non-empirical counterpart, namely, the population risk $R$.

A frequently used alternative to generalization bounds, which avoids the sampling error, are the *excess risk* bounds. That is, we are interested in upper bounding

$$R(w_n) - \inf_{w \in \mathcal{W}} R(w).$$

Via a standard decomposition, the generalization bounds of the form (1) can be translated into the excess risk bounds for the empirical risk minimization algorithm (ERM). However, in this case the sampling error $O(1/\sqrt{n})$ is propagated in the excess risk bound leading to suboptimal results in the cases where we expect the $O(1/n)$ rate of convergence. Thus, we are focusing on the following question:

*Can uniform stability provide high probability excess risk bounds with the rate (up to) $O(1/n)$?*

The main result of this paper answers this question positively and provides the first high probability bound based on uniform stability allowing the $O(\log n/n)$ rate of convergence. Similar questions appeared earlier in the literature on stochastic convex optimization, where optimal in-expectation results usually follow from stability. In particular, Shalev-Shwartz, Shamir, Srebro, and Sridharan asked in their pathbreaking paper [42] if a high probability excess risk bound for strongly convex and Lipschitz losses with the rate $O(1/n)$ is possible. As a corollary of our main result, we resolve their question by getting an almost optimal high probability bound with the rate $O(\log n/n)$.

## 1.1 Main results

It is well known that the $O(1/n)$ rate of convergence for the excess risk cannot be achieved for free. So we need to introduce an additional assumption. We consider the following generalization of the so-called *Bernstein condition* allowing multiple global risk minimizers. The version below is originally due to Koltchinskii [27].

**Assumption 1.1** (Generalized Bernstein condition). *Assume that $\mathcal{W}^* = \text{Argmin}_{w \in \mathcal{W}} R(w)$ is a set of risk minimizers in a closed set $\mathcal{W}$. We say that $\mathcal{W}$ together with the measure $P$ and the loss $\ell$ satisfy the generalized Bernstein assumption if for some $B > 0$ for any $w \in \mathcal{W}$, there is $w^* \in \mathcal{W}^*$ such that*

$$\mathbb{E}(\ell(w, Z) - \ell(w^*, Z))^2 \leq B(R(w) - R(w^*)).$$

Observe that the Bernstein assumption is independent of a specific learning algorithm. It is also not too restrictive and often accompanies uniform stability. In Section 2.1, we provide some examples and a detailed discussion.

Suppose that we are given a uniformly stable algorithm $w_n$ that attempts to minimize the empirical loss $R_n$. We denote the *optimization error* of such an algorithm by

$$\Delta_{\mathrm{opt}} = R_n(w_n) - \inf_{w \in \mathcal{W}} R_n(w).$$

In particular, for ERM, we have $\Delta_{\mathrm{opt}} = 0$. The following theorem is our first main result.

**Theorem 1.1.** *Assume that the loss $\ell(\cdot, \cdot)$ is bounded by $M$. Suppose also that Assumption 1.1 is satisfied with the parameter $B$. Let $w_n$ be a $\gamma$-stable algorithm that has the optimization error $\Delta_{\mathrm{opt}}$. There is an absolute constant $c > 0$ such that the following holds. Fix any $\eta > 0$. Then, with probability at least $1 - \delta$, it holds that*

$$R(w_n) - \inf_{w \in \mathcal{W}} R(w) \leq \Delta_{\mathrm{opt}} + \eta \mathbb{E} \Delta_{\mathrm{opt}} + c(1 + 1/\eta) \left( \gamma \log n + \frac{M + B}{n} \right) \log \left( \frac{1}{\delta} \right).$$

Our main application of this bound is an almost optimal high probability bound for ERM with strongly convex and Lipschitz losses. See Section 2.2 and Proposition 2.1 for more detail. Observe that the bound of Theorem 1.1 contains the term corresponding to the expected optimization error $\mathbb{E} \Delta_{\mathrm{opt}}$, where the expectation is taken with respect to the learning sample. This does not pose a problem in applications known to us. In particular, Theorem 1.1 implies that if our uniformly stable algorithm is ERM in $\mathcal{W}$, then $\Delta_{\mathrm{opt}} = 0$ and, with probability at least $1 - \delta$,

$$R(w_n) - \inf_{w \in \mathcal{W}} R(w) \leq c \left( \gamma \log n + \frac{M + B}{n} \right) \log \left( \frac{1}{\delta} \right).$$

Our second main result complements the generalization bound (1) and provides the *variance-type* bound, allowing us to completely remove the term $O(1/\sqrt{n})$ in (1) whenever the empirical error $R_n(w_n)$ is small.

**Theorem 1.2.** *There is an absolute constant $c > 0$ such that the following holds. Let $w_n$ be a $\gamma$-stable algorithm and assume that the loss $\ell(\cdot, \cdot)$ is bounded by $M$. Fix any $\eta > 0$. Then, with probability at least $1 - \delta$, it holds that*

$$R(w_n) \leq (1 + \eta) R_n(w_n) + c(1 + 1/\eta) \left( \gamma \log n + \frac{M}{n} \right) \log \left( \frac{1}{\delta} \right).$$

This result has a clear motivation: in modern practice, learning algorithms achieve a small or even zero empirical error on the learning sample, and the analysis should take this into account. Note that there are several recent variance-type stability bounds in the literature [32, 36] but under significantly stronger assumptions. In particular, in these papers, the loss is a generalized linear function, whereas we are working in the canonical framework of Bousquet and Elisseeff [7]. It is also important to note that in some examples, a small empirical error may lead to worse stability: this is called the *fitting-stability tradeoff* in the textbook [41]. For instance, in ridge regression, regularization improves stability but at the same time leads to an increased empirical error. And vice versa, by removing regularization, we may fit the data but lose stability.

**Additional notation.** For any two functions (or random variables) $f, g$ the symbol $f \lesssim g$ means that there is an absolute constant $c$ such that $f \leq cg$ on the entire domain. The gradient and subgradient of function $f$ at point $x_0$ are denoted by $\nabla f(x_0) = \nabla_x f(x_0)$ and $\partial f(x_0) = \partial_x f(x_0)$, respectively. The notation $\langle \cdot, \cdot \rangle$ stands for the inner product and by writing $\log x$, we conventionally mean $\max\{\log x, 1\}$.

## 2 Stochastic convex optimization with strongly convex losses

Stochastic convex optimization is a classical setup in which one minimizes a convex function $f$ based on some values or gradients at a given sequence of points. The most common setting is where at each round, the learner gets information on $f$ through a stochastic gradient oracle (see [39] and references therein). Another related setup that allows us to analyze generalization is when we observe the values of the losses $\ell(w, X_i)$ for an i.i.d. sample $X_1, \ldots, X_n$. Arguably the most well-studied case is when the following properties of the loss hold for any $x \in \mathcal{X}$:

- The loss $\ell(x, \cdot)$ is $\lambda$-strongly convex. That is, for any $w_1, w_2 \in \mathcal{W}, g \in \partial_w \ell(x, w_2)$,

$$\ell(x, w_1) - \ell(x, w_2) \geq \langle g, w_1 - w_2 \rangle + (\lambda/2)\|w_1 - w_2\|^2.$$

- The loss $\ell(x, \cdot)$ is $L$-Lipschitz. That is, for any $x \in \mathcal{X}$ and any $w_1, w_2 \in \mathcal{W}$,

$$|\ell(x, w_1) - \ell(x, w_2)| \leq L\|w_1 - w_2\|.$$

These assumptions on the loss are standard in the literature and have been studied in, e.g., [23, 24, 44, 42, 48, 49] as well as in the recent work on stability of gradient descent methods [20]. One can reasonably argue that both assumptions are rather restrictive (see the discussions in [44, 2]). Despite that, these assumptions are fundamental to the machine learning community and provide a clear illustration of our excess risk bounds.

In this setup, given a convex and closed set $\mathcal{W}$, we want to analyze the ERM strategy (also referred to as Sample Average Approximation (SAA)). That is, we are aiming to provide a high probability upper bound on the excess risk

$$R(\widehat{w}) - \inf_{w \in \mathcal{W}} R(w), \quad \text{where} \quad \widehat{w} = \operatorname{argmin}_{w \in \mathcal{W}} R_n(w).$$

The question of deviation optimal bounds in a closely related setup was recently revived by Harvey, Liaw, Plan, and Randhawa [21]. They proved a generalization of Freedman's inequality for martingale differences to show high probability guarantees for stochastic gradient descent, resolving several open questions. In our case, high probability excess risk bounds are known for some specific algorithms but follow from the regret bounds in the online setting combined with martingale-based online to batch conversion techniques [24] (see also [42, Section 2.2]). Despite numerous attempts [44, 43, 32, 48, 49], the question of whether dimension-free high probability bounds are achievable by *any* algorithm minimizing the empirical error remained open.

On the technical side, since ERM cannot be seen as a result of an online to batch conversion[1], the existing martingale-based techniques cannot be directly exploited. More importantly, uniform convergence, which is a standard tool for obtaining high probability bounds for ERM, fails in our case. This follows from an example in [43, Section 4.1 and page 2646] (see also [15]). One may wonder if a more precise localized analysis [34, 27, 3] should help in our setup. This is also not the case, since according to [42, Section 5.3] there is no uniform convergence for an arbitrary localization radius. Fortunately, our stability-based method proves the desired upper bound.

## 2.1 Verifying the Bernstein assumption

When applying Theorem 1.1, we first need to check that the Bernstein assumption holds. Let us discuss this assumption in more detail. Assumption 1.1 appears first in a similar generality in the work of Massart [34] and under the name *Bernstein class assumption* in [4]. This assumption is used as one of the components for proving the rates of convergence faster than $O\left(1/\sqrt{n}\right)$ (see the textbook [28]). The Bernstein assumption is usually implied by the convexity of the underlying class and the convexity of the loss function. We refer to [45] for an extensive survey on related results.

For our purposes, we verify Assumption 1.1 for strongly convex and Lipschitz losses. The following result is well-understood and appears (usually implicitly) in the literature. In our case, there is a unique risk minimizer $w^* \in \mathcal{W}$; that is, $w^* = \operatorname{argmin}_{w \in \mathcal{W}} R(w)$. From one perspective, the Lipschitz property implies for any $w \in \mathcal{W}$,

$$\mathbb{E}(\ell(w, X) - \ell(w^*, X))^2 \leq L^2\|w - w^*\|^2.$$

From another perspective, since the loss is $\gamma$-strongly convex and $w^*$ minimizes the risk in the convex set $\mathcal{W}$, we have

$$R(w) - R(w^*) \geq (\lambda/2)\|w - w^*\|^2.$$

Comparing the two inequalities, we have

$$\mathbb{E}(\ell(w, X) - \ell(w^*, X))^2 \leq L^2\|w - w^*\|^2 \leq (2L^2/\lambda)(R(w) - R(w^*)). \tag{2}$$

---

[1]The result in the textbook [11, Theorem 3.1] shows that the follow-the-leader strategy (an adaptation of ERM in the online setup) achieves the regret $4L^2(1 + \log n)/\lambda$ after $n$ rounds. However, after the online to batch conversion [24], we only get a high probability bound for an average of $n$ empirical risk minimizers.

This implies that $\lambda$-strongly convex and $L$-Lipschitz losses satisfy Assumption 1.1 with $B = 2L^2/\lambda$.

Our version of the Bernstein condition, namely Assumption 1.1, is due to Koltchinskii [27, Page 2618]. The key difference from the standard Bernstein assumption is that we allow multiple minimizers but can still provide $O(1/n)$ rates of convergence. Our motivation lies in the recent interest in relaxing the strong convexity assumption in (stochastic) optimization problems. One of such alternatives is the Polyak-Łojasiewicz condition (PL) (see [25]). In this context, the work [31] extends the standard Bernstein assumption to go beyond the strong convexity assumptions allowing multiple risk minimizers. Likewise, [12] claims that uniform stability results hold when the strong convexity assumption on the losses is replaced by the (PL) assumption[2]. Thus, our general results can potentially be useful in this direction.

## 2.2 High probability bound for almost risk minimizers

In this section, we present the main application of Theorem 1.1. In the strongly convex case, we provide a sharp high probability guarantee valid for *any* learning algorithm depending on its optimization error.

**Proposition 2.1.** *Let $\mathcal{W}$ be a convex closed set. Assume that the loss function is $\lambda$-strongly convex and $L$-Lipschitz as defined above. Let an approximate empirical minimizer $\widehat{w}$ have an optimization error $\Delta_{\mathrm{opt}}$ bounded by deterministic $\overline{\Delta}$ for any learning sample. Then, with probability $1 - \delta$,*

$$R(\widehat{w}) - R(w^*) \lesssim \overline{\Delta} + \left( \frac{L^2}{\lambda n} + \sqrt{\frac{L^2 \overline{\Delta}}{\lambda}} \right) \log n \log \left( \frac{1}{\delta} \right) .$$

*In particular, if $\widehat{w}$ is ERM in $\mathcal{W}$, then $\overline{\Delta} = 0$ and*

$$R(\widehat{w}) - R(w^*) \lesssim \frac{L^2}{\lambda n} \log n \log \left( \frac{1}{\delta} \right). \tag{3}$$

The in-expectation version of (3) without an additional $\log n$-factor is well-known and attributed to the foundational papers [7, 42, 43]. As we mentioned, the possibility of a high probability bound with the rate $O(1/n)$ was asked in [42, Discussion after Claim 6.2]. Despite the recent progress, the term $O(1/\sqrt{n})$ is present in the sharpest known high probability bound [17, Corollary 4.2]. Proposition 2.1 settles this question up to a logarithmic factor. We note that high-probability bounds are known for ERM in the particular case where the loss is a generalized linear function with a strongly convex penalty [44]. The analysis in [44] is based on localized Rademacher complexities and exploits the linear structure of the loss. As we mentioned above, uniform convergence cannot help in our setup.

## 2.3 Application to projected gradient descent without smoothness assumptions

Let us consider a simple illustration of Proposition 2.1. In what follows, we focus on the statistical rather than the computational part of the story. The method of *Projected Gradient Descent* (full-batch PGD) consists of iteration of the following update rules for $t = 1, \ldots, T$,

$$y_t = w_t - \nu_t g_t, \qquad \text{where } g_t \in \partial R_n(w_t),$$
$$w_{t+1} = \Pi_{\mathcal{W}}(y_t),$$

where $T$ is the total number of steps, $w_1$ is an initial approximation, and $\Pi_{\mathcal{W}}$ is the projection operator onto the convex closed set $\mathcal{W}$. The choice of the number of iterations $T$ and the step values $\nu_t$ affects the optimization error. For instance, when the loss is $\lambda$-strongly convex and $L$-Lipschitz, choosing $\nu_t = \frac{2}{\lambda(t+1)}$ gives the following optimization error (see [10, Theorem 3.9]),

$$R_n(\overline{w}_T) - \min_{w \in \mathcal{W}} R_n(w) \leq 4L^2/\lambda T,$$

where $\overline{w}_T = \frac{2}{T(T+1)} \sum_{t=1}^T t w_t$ is the weighted average of iterations. Therefore, PGD achieves the optimization error $O(1/n^2)$ after $T = O(n^2)$ steps. By Proposition 2.1, with probability at least $1 - \delta$, it holds that

$$R(\overline{w}_T) - R(w^*) \lesssim \frac{L^2}{\lambda n} \log n \log \left( \frac{1}{\delta} \right). \tag{4}$$

---

[2]The bounds in [12] require that (PL) holds for the empirical error. To the best of our knowledge, no stability results are known if (PL) is satisfied only for individual losses.

This is the first high probability $O(\log n/n)$ excess risk bound for non-smooth PGD. Our techniques do not give an answer to the question whether a smaller number of iterations is sufficient in the non-smooth case. Recent results suggest that full-batch PGD can indeed require significantly more steps than Stochastic Gradient Descent [1]. However, the authors only consider convex objectives that correspond to slow convergence rates $O(1/\sqrt{n})$.

We note that stability of PGD can be analyzed regardless of the optimization error. Indeed, the derivations of Hardt, Recht, and Singer [20, Section 3.4] (see also [17, Section 4.1.2]) imply that if the loss is $\beta$-smooth in addition to strong convexity and the Lipschitz property, that is,

$$\|\nabla_w \ell(x, w_1) - \nabla_w \ell(x, w_2)\| \leq \beta \|w_1 - w_2\|, \quad \text{for all } w_1, w_2 \in \mathcal{W},$$

then PGD with the constant step size $\nu = 1/\beta$ is $2L^2/(\lambda n)$-uniformly stable for any number of steps. In addition, the optimization error of the final iterate satisfies [10, Theorem 3.10],

$$R_n(w_T) - \min_{w \in \mathcal{W}} R_n(w) \lesssim \frac{\beta L^2}{\lambda^2} \exp\left(-\lambda T/\beta\right).$$

As a result, the smoothness assumption implies the same (previously unknown) high probability excess risk bound (4) after only $T = O(\log n)$ steps.

## 3 Proofs

Throughout the proofs, we rely on the $L_p$ norm. Denote the $L_p$-norm of a random variable $Z$ as $\|Z\|_p = (\mathbb{E}|Z|^p)^{1/p}$. A moment bound can be translated into a high-probability bound as follows (see, e.g., [9, Section 2]). Assume that for some $a, b > 0$ and all $p \geq 2$, it holds that $\|Z\|_p \leq a\sqrt{p} + bp$. Then, there is an absolute constant $C > 0$ such that for any $\delta \in (0, 1)$, with probability at least $1 - \delta$, it holds that

$$|Z| \leq C\left(a\sqrt{\log\left(1/\delta\right)} + b\log\left(1/\delta\right)\right). \tag{5}$$

As we mentioned, generalization bounds of form (1) cannot provide excess risk bounds with the rate better than $O(1/\sqrt{n})$. The following lemma separates the sampling term from the generalization error.

**Lemma 3.1.** *Let $w_n = w_n(X_1, \ldots, X_n)$ be a $\gamma$-stable algorithm and let $w'_n = w_n(X'_1, \ldots, X'_n)$ be its independent copy. Then, for any $p \geq 2$,*

$$\left\| R_n(w_n) - R(w_n) - \frac{1}{n}\sum_{i=1}^{n} \mathbb{E}[\ell(X_i, w'_n)|\ X_i] + \mathbb{E}R(w_n) \right\|_p \lesssim \gamma p \log n.$$

Such decomposition is possible due to the following extension of the bounded differences inequality by Bousquet, Klochkov, and Zhivotovskiy [9, Theorem 4].

**Theorem.** *Assume that $X_1, \ldots, X_n$ are independent variables and the functions $g_i : \mathcal{X}^n \to \mathbb{R}$ satisfy the following properties for $i = 1, \ldots, n$,*

- $\mathbb{E}_{X_i} g_i(X_1, \ldots, X_n) = 0$ *almost surely;*

- $g_i$ *has the bounded differences property with respect to all but the $i$-th variable: for all $j \neq i$ and $x_1, \ldots, x_n, x'_j$, we have $|g_i(x_1, \ldots, x_n) - g_i(x_1, \ldots, x_{j-1}, x'_j, x_{j+1}, \ldots, x_n)| \leq \beta$;*

- $|\mathbb{E}[g_i(X_1, \ldots, X_n)|\ X_i]| \leq K$ *almost surely.*

*Then, the following moment bounds hold for all $p \geq 2$,*

$$\left\|\sum\nolimits_{i=1}^{n} g_i\right\|_p \leq 12\sqrt{2}\beta pn \log n + 4K\sqrt{pn}. \tag{6}$$

In addition, we will use the following version of the Bernstein inequality [6, Theorem 15.11]: if $X_1, \ldots, X_n$ are zero mean, independent and bounded $|X_i| \leq M$ almost surely, then

$$\|X_1 + \cdots + X_n\|_p \leq 6\sqrt{\left(\sum\nolimits_{i=1}^{n} \mathbb{E}X_i^2\right)p} + 4pM, \qquad \forall p \geq 2. \tag{7}$$

Our last tool is the concentration inequality for non-negative *weakly self-bounded* functions. Assume that $a, b \geq 0$. We say that the function $f : \mathcal{X}^n \to [0, +\infty)$ if $(a, b)$-weakly self-bounded if there exist functions $f_i : \mathcal{X}^{n-1} \to [0, +\infty)$ that satisfy for all $x \in \mathcal{X}^n$,

$$\sum_{i=1}^n (f(x) - f_i(x))^2 \leq af(x) + b.$$

The following concentration inequality is a lower tail version of [6, Theorem 6.19], which is originally due to Maurer [35]. The difference is that in their result it is assumed that $f_i(x) \leq f(x)$ for any $x \in \mathcal{X}^n$. The proof of the version below is standard, and we reproduce it below for the sake of completeness. Since we consider the lower tail, we remove the term $at$ present in [6, Theorem 6.19].

**Proposition 3.1.** *Suppose that $X_1, \ldots, X_n$ are independent random variables and the function $f : \mathcal{X}^n \to [0, +\infty)$ is $(a, b)$-weakly self-bounded, and the corresponding functions $f_i$ satisfy $f_i(x) \geq f(x)$ for $i = 1, \ldots, n$ and any $x \in \mathcal{X}^n$. Then, for any $t > 0$,*

$$\Pr(\mathbb{E}f(X_1, \ldots, X_n) \geq f(X_1, \ldots, X_n) + t) \leq \exp\left(-\frac{t^2}{2a\mathbb{E}f(X_1, \ldots, X_n) + 2b}\right). \quad (8)$$

## 3.1 Proof of Lemma 3.1

For $w_n^{(i)} = w_n(X_1, \ldots, X_{i-1}, X_i', X_{i+1}, \ldots, X_n)$, where $X_i'$ is an independent copy of $X_i$, consider the functions

$$g_i(X_1, \ldots, X_n) = \mathbb{E}_{X_i'}\ell(X_i, w_n^{(i)}) - \mathbb{E}_{X_i'}R(w_n^{(i)}).$$

One can immediately verify that these functions satisfy all three properties needed to apply (6) with $\beta = 2\gamma$. It is standard to check that (see e.g., [9, Lemma 7])

$$\left|n(R_n(w_n) - R(w_n)) - \sum_{i=1}^n g_i\right| \leq 2\gamma n.$$

Let us consider for $i = 1, \ldots, n$, the functions $h_i(X_1, \ldots, X_n) = g_i - \mathbb{E}[g_i | X_i]$, where the functions $h_i$ preserve the stability property (up to a factor of 2). Observe that $\mathbb{E}[h_i | X_i] = 0$ almost surely, which implies $K = 0$. Therefore, applying (6) to the functions $h_i$, we have that for any $p \geq 2$,

$$\left\|\sum_{i=1}^n g_i - \mathbb{E}[g_i | X_i]\right\|_p \leq 48\sqrt{2}\gamma pn \log n.$$

Notice that $\mathbb{E}[g_i | X_i] = \mathbb{E}[\ell(X_i, w_n') | X_i] - \mathbb{E}R(w_n')$. Our result follows. $\qquad \square$

## 3.2 Proof of Theorem 1.1

The proof starts with a standard decomposition that turns the generalization bound into an excess risk bound. Denote $R^* = \inf_{w \in \mathcal{W}} R(w)$. We have for any $w^* \in \mathrm{Argmin}_{w \in \mathcal{W}} R(w)$,

$$R(w_n) - R^* = R(w_n) - R_n(w_n) + R_n(w_n) - R_n(w^*) + R_n(w^*) - R^*$$
$$\leq \Delta_{\mathrm{opt}} - (R_n(w_n) - R(w_n)) + R_n(w^*) - R^*.$$

Here, the expression $R_n(w_n) - R(w_n)$ stands for the generalization error and is typically of order $1/\sqrt{n}$. To avoid this, we use the decomposition of Lemma 3.1,

$$R_n(w_n) - R(w_n) = \xi + \frac{1}{n}\sum_{i=1}^n \mathbb{E}'\ell(X_i, w_n') - \mathbb{E}R(w_n),$$

where $\|\xi\|_p \lesssim \gamma p \log n$ for any $p \geq 2$ and $w_n'$ is an independent copy of $w_n$. We write $\mathbb{E}'$ to denote the expectation with respect to this independent copy. We now need to pair the remainder term with $R_n(w^*) - R(w^*)$ to achieve the $O(1/n)$ rate. Since $\mathbb{E}R(w_n) = \mathbb{E}'R(w_n')$, then for any $w^* \in \mathrm{Argmin}_{w \in \mathcal{W}} R(w)$, it holds that

$$R(w_n) - R^* \leq \Delta_{\mathrm{opt}} - \xi - \frac{1}{n}\sum_{i=1}^n (\mathbb{E}'\ell(X_i, w_n') - \ell(X_i, w^*)) + \mathbb{E}'R(w_n') - R^*.$$

Since we are free to choose any $w^*$, let us take the one corresponding to $w_n'$ in Assumption 1.1. Notice that neither $R(w_n)$, $\xi$ nor $\Delta_{\mathrm{opt}}$ depend on this choice. In other words, $w' \in \mathrm{Argmin}_{w \in \mathcal{W}} R(w)$

is a random vector induced by $w'_n$, where we write $w'$ instead of $w^*$ to point out this dependence. Therefore, we rewrite our last display as follows

$$R(w_n) - R^* \leq \Delta_{\text{opt}} - \xi - \frac{1}{n} \sum_{i=1}^{n} \left( \mathbb{E}' \ell(X_i, w'_n) - \ell(X_i, w') \right) + \mathbb{E}' R(w'_n) - R^* \,.$$

Notice that here the only terms that depend on $w'_n$ are $\ell(X_i, w')$. Taking the expectation $\mathbb{E}'$ of both sides of this inequality, we obtain

$$R(w_n) - R^* \leq \Delta_{\text{opt}} - \xi - \frac{1}{n} \sum_{i=1}^{n} \mathbb{E}' \left[ \ell(X_i, w'_n) - \ell(X_i, w') \right] + \mathbb{E}' R(w'_n) - R^* \,. \tag{9}$$

Here, $\mathbb{E}\mathbb{E}' \ell(X_i, w') = \mathbb{E}'\mathbb{E}[\ell(X_i, w') | w'_n] = \mathbb{E}' R(w') = R^*$, and as we have already noticed, $\mathbb{E}\mathbb{E}' \ell(X_i, w'_n) = \mathbb{E}' R(w'_n)$. Moreover, by the Bernstein condition and Jensen's inequality,

$$
\begin{aligned}
\mathbb{E} \left( \mathbb{E}'[\ell(X_i, w'_n) - \ell(X_i, w')] \right)^2 &\leq \mathbb{E}'\mathbb{E}(\ell(X_i, w'_n) - \ell(X_i, w'))^2 \\
&= \mathbb{E}'\mathbb{E}[(\ell(X_i, w'_n) - \ell(X_i, w'))^2 | w'_n] \\
&\leq B(\mathbb{E}' R(w'_n) - R^*) \,.
\end{aligned}
$$

Having this variance bound, we are ready to apply the moment Bernstein inequality (7) to the sum of independent random variables $\mathbb{E}'[\ell(X_i, w'_n) - \ell(X_i, w')]$. Since $\mathbb{E}' R(w'_n) - R^*$ is exactly the expectation of each of these summands, we have for all $p \geq 2$,

$$\left\| \frac{1}{n} \sum_{i=1}^{n} \mathbb{E}' \left[ \ell(X_i, w'_n) - \ell(X_i, w') \right] - \mathbb{E}' R(w'_n) + R^* \right\|_p \lesssim \sqrt{B(\mathbb{E}R(w_n) - R^*) \frac{p}{n}} + \frac{pM}{n} \,. \tag{10}$$

Plugging this into (9), we obtain for each $p \geq 2$ and some absolute constant $C > 0$,

$$
\begin{aligned}
\| R(w_n) - R^* - \Delta_{\text{opt}} \|_p &\leq C \left( \gamma p \log n + \sqrt{B(\mathbb{E}R(w_n) - R^*) \frac{p}{n}} + \frac{pM}{n} \right) \\
&\leq \eta(\mathbb{E}R(w_n) - R^*) + C \left( \gamma p \log n + \left( \frac{B}{\eta} + M \right) \frac{p}{n} \right) \,, \tag{11}
\end{aligned}
$$

where the second inequality holds since for any $a, b, \eta > 0$, it holds that $\sqrt{ab} \leq \eta a + b/\eta$.

Finally, we need an upper bound on $\mathbb{E}R(w_n) - R^*$. Taking $p = 2$ in (11) and using the Cauchy-Schwarz inequality, we have

$$
\begin{aligned}
\mathbb{E}R(w_n) - R^* - \mathbb{E}\Delta_{\text{opt}} &\leq \| R(w_n) - R^* - \Delta_{\text{opt}} \|_2 \\
&\leq \eta(\mathbb{E}R(w_n) - R^*) + C \left( 2\gamma \log n + 2(B/\eta + M)/n \right) \,.
\end{aligned}
$$

Subtracting $\eta(\mathbb{E}R(w_n) - R^*)$ from both sides and dividing by $1 - \eta$, we obtain

$$\mathbb{E}R(w_n) - R^* \leq \frac{1}{1-\eta} \mathbb{E}\Delta_{\text{opt}} + \frac{C}{1-\eta} \left( \gamma \log n + \left( \frac{B}{\eta} + M \right) \frac{1}{n} \right) \,.$$

Plugging this bound back into (11), assuming that $\eta < 1/2$, and translating the moment bound into the high-probability bound through (5), we obtain that, with probability at least $1 - \delta$,

$$R(w_n) - R^* \leq \Delta_{\text{opt}} + C' \left( \frac{\eta}{1-\eta} \mathbb{E}\Delta_{\text{opt}} + \gamma \log n \log \left( \frac{1}{\delta} \right) + \left( \frac{B}{\eta} + M \right) \frac{\log(1/\delta)}{n} \right) \,,$$

where $C' > 0$ is an absolute constant. By replacing $\eta$ by $\frac{\eta}{\max\{C', 2\}(1+\eta)}$, we finish the proof. $\square$

### 3.3 Proof of Theorem 1.2

We will show that under the conditions of the theorem, the following variance bound holds. For any $\delta \in (0, 1)$, we have, with probability at least $1 - \delta$,

$$R(w_n) - R_n(w_n) \lesssim \gamma \log n \log \left( \frac{1}{\delta} \right) + \sqrt{\frac{MR(w_n)}{n} \log \left( \frac{1}{\delta} \right)} + \frac{M}{n} \log \left( \frac{1}{\delta} \right) \,. \tag{12}$$

The statement of the theorem follows immediately by applying the inequality $\sqrt{ab} \le \eta a + b/\eta$ to the middle term of the right-hand side and choosing the appropriate value of $\eta$.

The proof of (12) repeats the arguments of Theorem 1.1 with several important changes listed below. As in the proof of Theorem 1.1, we use the generalization bound of Lemma 3.1, and then apply the Bernstein inequality to the correcting term. Converting the moment bound into a high probability bound by (5), we have, with probability $1 - \delta/2$,

$$R(w_n) - R_n(w_n) \lesssim \gamma \log n \log\left(\frac{1}{\delta}\right) + \sqrt{\frac{\mathbb{E}(\ell(X', w_n))^2}{n} \log\left(\frac{1}{\delta}\right)} + \frac{M}{n} \log\left(\frac{1}{\delta}\right), \qquad (13)$$

where we used that the variance of $\mathbb{E}[\ell(X, w'_n)| X]$ is bounded by $\mathbb{E}(\ell(X', w_n))^2$ due to Jensen's inequality.

Our goal is to replace the non-random term $\mathbb{E}(\ell(X', w_n))^2$ with its empirical version $\mathbb{E}'(\ell(X', w_n))^2$, where slightly abusing the notation, $\mathbb{E}'$ denotes the integration only with respect to the independent copy $X'$. Unfortunately, a naive application of the bounded difference inequality leads to a suboptimal bound in our case. Instead, we use second order concentration through the weakly self-bounding property. Set

$$f = f(x_1, \ldots, x_n) = \mathbb{E}'(\ell(X', w_n(x_1, \ldots, x_n)))^2$$

and $f_i = \sup_{x_i \in \mathcal{X}} f(x_1, \ldots, x_n)$, so that $f_i \ge f$ for all $i = 1, \ldots, n$. We show that $f$ is $(8n\gamma^2, 2n\gamma^4)$-weakly self-bounded. By the uniform stability and Jensen's inequality, we have

$$\sum_{i=1}^{n} (f - f_i)^2 \le \sum_{i=1}^{n} (\mathbb{E}'(\ell(X', w_n))^2 - \sup_{x_i \in \mathcal{X}} \mathbb{E}'(\ell(X', w_n))^2)^2$$
$$\le n\gamma^2 (2\mathbb{E}'\ell(X', w_n) + \gamma)^2$$
$$\le 8n\gamma^2 f + 2n\gamma^4.$$

Therefore, by the concentration inequality (8) we have that, with probability $1 - \delta/2$,

$$\mathbb{E}(\ell(X', w_n))^2 - \mathbb{E}'(\ell(X', w_n))^2 \lesssim \sqrt{(n\gamma^2 \mathbb{E}(\ell(X', w_n))^2 + n\gamma^4) \log(1/\delta)}.$$

Using $\sqrt{ab} \le a + b$ for all $a, b \ge 0$ and $\mathbb{E}'(\ell(X', w_n))^2 \le MR(w_n)$, we obtain on the same event

$$\mathbb{E}(\ell(X', w_n))^2 - 2MR(w_n) \lesssim n\gamma^2 \log(1/\delta).$$

Plugging this bound into (13) and using the union bound, we obtain (12). Hence, the theorem follows. □

## 3.4 Proof of Proposition 2.1

We first check the uniform stability of $\widehat{w}$. For this we need to prove that for any $x \in \mathcal{X}$,

$$|\ell(x, \widehat{w}) - \ell(x, \widehat{w}^{(i)})| \le 4L^2/(\lambda n) + \sqrt{8L^2 \overline{\Delta}/\lambda},$$

where $\widehat{w} = \widehat{w}(x_1, \ldots, x_n)$ and $\widehat{w}^{(i)} = \widehat{w}(x_1, \ldots, x_{i-1}, x'_i, x_{i+1}, \ldots, x_n)$. Let also $\widetilde{w}$ be the minimizer of $R_n$, which denotes the empirical risk on the sample $x_1, \ldots, x_n$, and $\widetilde{w}^{(i)}$ is the minimizer of $R_n^{(i)}$, which denotes the empirical risk on the sample $x_1, \ldots, x'_i, \ldots, x_n$. Then, by [42, Claim 6.1],

$$\text{for any } x \in \mathcal{X}, \qquad |\ell(x, \widetilde{w}) - \ell(x, \widetilde{w}^{(i)})| \le 4L^2/(\lambda n).$$

On the other hand, since $R_n$ is $\lambda$-strongly convex,

$$(\lambda/2)\|\widehat{w} - \widetilde{w}\|^2 \le R_n(\widehat{w}) - R_n(\widetilde{w}) \le \overline{\Delta},$$

which implies $\|\widehat{w} - \widetilde{w}\| \le \sqrt{2\overline{\Delta}/\lambda}$. A similar relation holds between $\widehat{w}^{(i)}$ and $\widetilde{w}^{(i)}$. Using the $L$-Lipschitz property, we conclude that for all $x$,

$$|\ell(x, \widehat{w}) - \ell(x, \widehat{w}^{(i)})| \le |\ell(x, \widetilde{w}) - \ell(x, \widetilde{w}^{(i)})| + |\ell(x, \widetilde{w}^{(i)}) - \ell(x, \widehat{w}^{(i)})| + |\ell(x, \widehat{w}) - \ell(x, \widetilde{w})|$$
$$\le 4L^2/(\lambda n) + \sqrt{8L^2 \overline{\Delta}/\lambda}.$$

Since $\widehat{w}$ is stable, we apply Theorem 1.1. It is only left to check that the loss is bounded. This follows from the fact that it is both $L$-Lipschitz and $\lambda$-strongly convex at the same time. Indeed, we have for any $w \in \mathcal{W}$ and $w^* = \operatorname{argmin}_{w \in \mathcal{W}} R(w)$, that

$$(\lambda/2)\|w - w^*\|^2 \leq R(w) - R(w^*) \leq L\|w - w^*\|,$$

so that the convex set $\mathcal{W}$ is bounded and contained in the ball $\{w : \|w - w^*\| \leq 2L/\lambda\}$. Using again the Lipschitz property of $\ell(x, \cdot)$ we conclude that for any $x \in \mathcal{X}, w \in \mathcal{W}$,

$$|\ell(x, w) - \ell(x, w^*)| \leq 2L^2/\lambda.$$

Although the conditions of Theorem 1.1 require a uniform bound $\ell(\cdot, \cdot) \leq M$, it only enters in the proof in (10), where we apply the Bernstein inequality (7) to the sum of independent random variables $\mathbb{E}[\ell(X_i, w'_n) - \ell(X_i, w^*)| \, X_i]$. Therefore, the inequality still holds with $2L^2/\lambda$ in place of $M$. The rest of the proof of Theorem 1.1 provides us with the required bound. $\qquad\square$

### 3.5 Proof of Proposition 3.1

Since the result is not presented in the literature in the form we need, we reproduce the standard argument. Let $Z = f(X_1, \ldots, X_n)$, $Z_i = f_i(X_1, \ldots, X_{i-1}, X_{i+1}, \ldots, X_n)$ are such that $Z \leq Z_i$ almost surely, and the weakly self-bounding property holds, that is, $\sum_{i=1}^n (Z_i - Z)^2 \leq aZ + b$. Let us first apply a modified logarithmic Sobolev inequality [6, Theorem 6.6]. We have

$$\lambda \mathbb{E}[Ze^{\lambda Z}] - \mathbb{E}[e^{\lambda Z}] \log \mathbb{E}[e^{\lambda Z}] \leq \mathbb{E}\left(e^{\lambda Z} \sum_{i=1}^n \phi(-\lambda(Z - Z_i))\right),$$

where $\phi(x) = e^x - x - 1$. Since for $x \geq 0$, $\phi(-x) \leq x^2/2$ and $Z - Z_i \leq 0$ for all $i = 1, \ldots, n$, we have for any $\lambda \leq 0$,

$$\lambda \mathbb{E}[Ze^{\lambda Z}] - \mathbb{E}[e^{\lambda Z}] \log \mathbb{E}[e^{\lambda Z}] \leq \frac{\lambda^2}{2}\mathbb{E}\left(e^{\lambda Z} \sum_{i=1}^n (Z - Z_i)^2\right) \leq \frac{\lambda^2}{2}(a\mathbb{E}[Ze^{\lambda Z}] + b\mathbb{E}[e^{\lambda Z}]).$$

Define $G(\lambda) = \log \mathbb{E}[e^{\lambda Z}]$, so that $G'(\lambda) = \mathbb{E}[Ze^{\lambda Z}]/\mathbb{E}[e^{\lambda Z}]$. Dividing both sides of the last display by $\mathbb{E}[e^{\lambda Z}]$, we obtain

$$\lambda G'(\lambda) - G(\lambda) \leq \frac{\lambda^2}{2}(aG'(\lambda) + b), \qquad \lambda \leq 0.$$

For $\lambda < 0$, we have

$$\left(\left(\frac{1}{\lambda} - \frac{a}{2}\right)G(\lambda)\right)' = \left(\frac{1}{\lambda} - \frac{a}{2}\right)G'(\lambda) - \frac{G(\lambda)}{\lambda^2} \leq \frac{b}{2}.$$

Finally, we integrate this inequality. Observe that $G(0) = 0$ and we also have $G'(0) = \mathbb{E}Z$. Hence, as $\lambda \to -0$, by Taylor's theorem $(1/\lambda - a/2)G(\lambda) = (1/\lambda - a/2)(\lambda \mathbb{E}Z + o(\lambda)) = \mathbb{E}Z + o(1)$. Integrating the last display over the interval $[\lambda, 0]$, we obtain

$$\mathbb{E}Z - \left(\frac{1}{\lambda} - \frac{a}{2}\right)G(\lambda) \leq (-\lambda)\frac{b}{2}.$$

After some rearrangements this leads to the following inequality (recall that $\lambda < 0$),

$$\log \mathbb{E}[e^{\lambda(Z - \mathbb{E}Z)}] = G(\lambda) - \lambda \mathbb{E}Z \leq \frac{\lambda^2(a\mathbb{E}Z + b)}{2(1 - \lambda a/2)} \leq \frac{\lambda^2}{2}(a\mathbb{E}Z + b).$$

It remains to use the Markov inequality (recall again that $\lambda < 0$) to show that

$$\mathbb{P}(Z < \mathbb{E}Z - t) = \mathbb{P}\left(\lambda(Z - \mathbb{E}Z) > -\lambda t\right) = \mathbb{P}\left(e^{\lambda(Z - \mathbb{E}Z + t)} > 1\right) \leq \exp\left(t\lambda + \frac{\lambda^2}{2}(a\mathbb{E}Z + b)\right).$$

The latter exponent is minimized by $\lambda = -t/(a\mathbb{E}Z + b) < 0$ implying the statement. $\qquad\square$

**Acknowledgments.** We thank Jaouad Mourtada for his comment on the follow-the-leader strategy and for providing several important references. We also thank Tomas Vaškevičius for valuable feedback. Nikita Zhivotovskiy is funded in part by ETH Foundations of Data Science (ETH-FDS).

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
