# OpenReview forum: "Stability and Deviation Optimal Risk Bounds with Convergence Rate $O(1/n)$"
_NeurIPS.cc/2021/Conference — NeurIPS 2021 Oral_

### Official Review · Reviewer_1bRR · 2021-07-07

**Rating:** 7
**Confidence:** 5

**Summary:**

This work provides improved generalization bounds for uniformly stable algorithms under the Bernstein condition. This condition bounds the variance of the loss with respect to a risk minimizer by the excess risk, and it has been widely used to derive faster rates in statistical learning. However, their use for high probability bounds in stochastic optimization was limited by the fact that all known high probability generalization bounds for stable algorithms incurred terms $\Omega(1/\sqrt{n})$. The main consequence of this result is a high probability excess risk bound for the Empirical Risk Minimizer in strongly convex stochastic optimization, resolving an open question by Shalev-Shwartz et al. (COLT 2009). Another consequence is an (optimal) high probability excess risk bound for (full-batch) gradient descent with O(n^2) iterations.

Uniform stability is a central tool for machine learning theory, and the results of this paper provide a clear contribution here. Technically, the paper does not include much innovation, but the methods used, particularly referring to the excess risk decomposition, are quite interesting, and may encounter further use. Finally, I am not sure about the "deviation optimal" part in the title: Is it clear that the bounds derived are sharp? No matching lower bounds are provided for Lemma 3.1 or Theorem 1.1. I believe existing results (like lower bounds in Bousquet, Klochkov & Zhivotovskiy (BKZ)) may already explain this to some extent, but I would suggest including a more detailed discussion on sharpness.


**Ethical Concerns:**

No ethical concerns from this paper.

**Limitations And Societal Impact:**

This work is theoretical, so I agree with the authors there are no societal consequences of it.

**Main Review:**


1. To achieve the main result, a useful Lemma 3.1 is shown (this is a minor variation of a Lemma from BKZ): this result avoids the sampling error term $O(1/\sqrt{n})$ by showing a moment bound of a "shifted" generalization error. This shifting includes the risk of the algorithm output and an average conditional loss. It would be interesting if the authors would elaborate further about the intuition on this refinement, and what are its consequences for the excess risk bounds (the technical proofs are clear, but
nevertheless it might be useful a brief discussion).

2. The proof of the main result (Thm 1.1) is obtained nontrivially from Lemma 3.1. This involves a (seemingly novel) decomposition of the excess risk and the (shifted) generalization error, along with various estimates based on concentration by the Bernstein condition.

3. Theorem 1.2 provides what they call a variance-type bound (not sure why is it called like this), which bounds the risk in terms of the empirical risk and a deviation bound which leverages faster rates by uniform stability. This bound seems useful in realizable settings, where the empirical risk can be made arbitrarily small.

4. The consequences for strongly convex stochastic optimization appear in Proposition 2.1 and its following discussion. As mentioned above, this general bound for stable algorithms answers a question raised in Shalev-Shwartz et al (2009) which is quite interesting on its own. In this result, the most problematic term is $\sqrt{L^2\overline{\Delta}/\lambda}$, where $\overline{\Delta}$ is a bound on the empirical error: this leads to optimal excess risk in this case for GD with O(n^2) iterations. It is worth pointing out here that a similar iteration requirement appears in the nonsmooth case, as shown in Bassily, Feldman Guzman and Talwar (2020). That paper also includes a lower bound on stability, showing that for GD $\Omega(n^2)$ steps needed to obtain optimal rates with uniform stability. It is reasonable to conjecture that a similar lower bound may hold in the strongly convex setting, and in fact some limitations of GD in terms of excess risk have already been shown in recent work by Amir, Koren and Livni (https://arxiv.org/pdf/2102.01117.pdf). The latter result is shown for GD on the regularized ERM, but I suspect ideas there might be useful for the strongly convex setting as well.

Finally, I would suggest the authors to include a more detailed statement of the last paragraph before section 3 (lines 204-206 in page 6). Here it is claimed a new high-probability excess risk bound for GD on smooth and strongly convex losses. For future use of this result, it might be a good idea to have a proper statement.


**Time Spent Reviewing:**

4

---

> ### Author Response · Authors · 2021-08-09
> **Answer to the reviewer**
>
> "Deviation optimal" refers mainly to the bound from Proposition 2.1, although rather ambitiously we ignore the logarithmic term. The lower bound $\Omega(\log(1/\delta)/n)$ is a standard excess risk lower bound for strongly convex losses (one may consider $d = 1, X = 1$ a.s., then $(y - w)^2$ is strongly convex with respect to $w$ whenever $Y$ is equal to $\pm L$ with probability $1/2$ respectively. In this case, ERM is a sample mean estimator and its excess risk is $\Omega(\log(1/\delta)/n)$. We will add a comment and a reference on this. Lemma 3.1 as a probabilistic statement that is also sharp up to a logarithmic factor as the lower bound of Proposition 9 in (BKZ) shows.
>
> We call it the variance-type bound because of the equivalent form presented in equation (12). The second moment of the risk is bounded there by the term $MR(w_n)$. This notion is quite standard for the bounds based on the Bernstein inequality.
>
> Thanks a lot, we were not aware of the recent work of Amir, Koren and Livni (2021) and some of the details of Bassily, Feldman, Guzman and Talwar (2020). We will add this discussion and references in the updated text. It is indeed very plausible that $\Omega(n^2)$ steps are needed in the strongly convex, non-smooth case.
>
> Thank you for this suggestion, we will expand the discussion of the strongly convex smooth case in the final version.

---

> > ### Comment · Reviewer_1bRR · 2021-08-13
> > **Answer to authors**
> >
> > Thank you for the clarifications.

---

### Official Review · Reviewer_3CZg · 2021-07-14

**Rating:** 7
**Confidence:** 5

**Summary:**

 The authors presented an improved variance-type excess risk analysis for uniformly stable learning algorithms. Particularly, it was proved that if the population risk satisfies the Bernstein condition, then a $\gamma$-stable algorithm with optimization error $\Delta_{opt}$ over a set of $n$ observations can achieve excess risk of order $\gamma \log(n) + \frac{1}{n} + \Delta_{opt}$ with high probability. This result implies $O(\frac{\log(n)}{\lambda n})$ risk bounds for empirical risk minimization (ERM) and projected gradient descent (PGD) with $\lambda$-strongly convex and Lipschitz loss functions. Also, a high-probability $\eta$-approximated generalization error bound is provided which improves the $O(\sqrt{\frac{\log(1/\delta)}{n}})$ tail term in the best known generalization bounds to $O(\frac{\log(1/\delta)}{\eta n})$.



**Limitations And Societal Impact:**

Yes,  the authors have partially addressed the limitations and potential negative societal impact of their work.

**Main Review:**

Strengths:
+ The deviation analysis of uniformly stable algorithms as focused in this paper is an important and timely topic in learning theory.

+ I am particularly impressed with the strength and originality of the results obtained in this paper for strongly convex optimization problems. The main idea of proof is to make a smart use of an extended bounded difference inequality from Bousquet, et al. (2020) to separate the sampling term from the generalization error, so that the randomness of model and data sample can be decoupled in a way that the Bernstain inequality is applicable to get tighter bounds. While built largely upon the previous results, IMO the technical part still appears novel enough in the considered problem regime.

+ The paper is generally well organized and clearly presented, with all steps in the proofs very carefully explained.

Weakness:

   My main concern is about the benefit of main results in Theorem 1.1 when substantialized to ERM and PGD with arbitrary convex losses. Basically, the improved $O(\frac{\log(n)}{n})$ rate of convergence seems only possible for strongly convex loss functions with zero optimization error achievable over training data. For generic convex losses which are of more interest in practice, we usually need to deal with regularized ERM of which the empirical risk error $\Delta_{opt}$ is typically $O(\frac{1}{\sqrt{n}})$ due to a trade-off between optimization and generalization. In this case, while the population risk can still be assumed strongly convex so that Theorem 1.1 is applicable, the $O(\frac{1}{\sqrt{n}})$ optimization error term will dominate the risk bound which makes the  $O(\frac{1}{n})$ term much less interesting. The authors have partially  addressed such a limitation for Theorem 1.2 with respect to the empirical risk. However, it looks like that similar or even more severe limitations exist for Theorem 1.1 as well in terms of optimization error.


=== Post rebuttal update ===

The major concern  on the limitation of results raised in my initial review has been properly addressed in author response. I would like to maintain my rating.

**Time Spent Reviewing:**

8

---

> ### Author Response · Authors · 2021-08-09
> **Answer to the reviewer**
>
> Thank you for the detailed review. General convex losses are indeed of greater practical interest. However, we respectfully disagree that the fact that our improvements are not very useful in this setup is the weakness of our work. In particular, there are fundamental limitations for achieving the rates faster than $O(1/\sqrt n)$ for general convex Lipschitz losses (for some high probability bounds achieving this rate, see the work of Feldman and Vondrak, 2019), whereas we focus on getting $O(1/n)$ high probability bounds whenever it is theoretically possible. We also remark that for strongly convex losses the improved convergence rates work if the optimization error is small but not necessarily zero (see Section 2.3). Finally, we briefly discuss possible applications in the non-strongly convex cases (see last paragraph of Section 2.1).

---

> > ### Comment · Reviewer_3CZg · 2021-08-18
> > **Possibility of removing the logarithmic factors**
> >
> > Thank you for the feedback. My concern on the limitation of results is completely clarified. I would maintain my rating and provide an additional comment on the possibility of removing the $\log(n)$ factor in the risk bound for $\lambda$-strongly convex ERM. I note that this seems possible if the loss function is additionally assumed to be smooth. In this case, the $\gamma$-stability of $w_n$ implies that of $\|\|w_{n} – w^* \|\|$. Then we have
> >
> > $E[\|\|w_n – w^* \|\|] \le \sqrt{E[\|\|w_n – w^* \|\|^2]} \le \sqrt{E\left[\frac{2(R(w^n) – R(w^*))}{\lambda}\right]} \le O(\sqrt{\frac{\gamma}{\lambda}})$
> >
> > Applying McDiamid’s inequality yields that $\|\|w_n – w^* \|\| \le O\left(\sqrt{\frac{\gamma}{\lambda}} + \gamma\sqrt{n\log(1/\delta)}\right)$ holds with probability at least $1-\delta$. The smoothness the leads to $R(w_n) – R(w^*) \le O(\|\|w_n – w^*\|\|^2) \le O(\frac{\gamma}{\lambda} + \gamma^2 n\log(1/\delta)) \le O(\frac{\log(1/\delta)}{\lambda^2 n} ) $, keeping in mind that $\gamma\le O(\frac{1}{\lambda n})$.
> >
> > I’m wondering if the $\log(n)$ factor can also be improved for Lipschitz losses, at least for strongly convex ERM when $\gamma$ scales as fast as  $O(1/n)$?

---

> > > ### Author Response · Authors · 2021-08-21
> > > **Re: Possibility of removing the logarithmic factors**
> > >
> > > The question of log factor is very thrilling but so far the technique we use does not allow us to get rid of it.
> > >
> > > Thank you for this interesting and short proof. It seems however that such an approach is limited to $\lambda$ of order one since the term $1/(\lambda^{2} n)$ can be worse than ours $1/ (\lambda n) $ when $\lambda$ is somewhere between $1/\sqrt{n}$ and $1$.

---

### Official Review · Reviewer_KWWB · 2021-07-14

**Rating:** 9
**Confidence:** 4

**Summary:**

The article exhibit a way to bound the excess risk in a general risk minimization framework when the loss function is with bounded difference (or stable as they call it). They show that the rates $O(1/n)$ can be attained under some Bernstein condition. This is  often called "fast rates" of convergence in the literature and this article exhibit the first general inequality to get fast rates of convergence for stable loss function.

**Main Review:**

To my knowledge, this type of result was only proved in particular cases and this is the first general result on fast rates of convergence obtained via stability. The article is clear and well written and I think this is significant for the Neurips community. This article may generalize to a lot of learning problems and may become a tool to prove fast rates in  a lot of different settings, this may be an important step to assess the rate of convergence of algorithms and to have theoretical rates that match the rates that we witness in practice. The authors also illustrate this generalization capacity on the problem of stochastic convex optimization with strongly convex losses.

**Time Spent Reviewing:**

2

---

### Official Review · Reviewer_Bxnk · 2021-07-16

**Rating:** 7
**Confidence:** 4

**Summary:**

This paper presents high probability bounds of O(1/n) for uniformly stable algorithms. The authors assume the generalized Bernstein condition due to Koltchinskii [26] as an additional assumption compared to related works in the literature. Under this assumption, they show that uniformly stable algorithms can achieve [optimization_error + O(1/n)] excess risk bound with a high probability. The authors show that optimization of stronlgy convex Lipschitz continuous objective functions satisfies the generalized Bernstein condition. This allows them to provide the first high probability O(log(n)/ n) risk bound for uniformly stable algorithms on a strongly convex Lipschitz continuous optimization problem.

**Limitations And Societal Impact:**

One potential improvement is to consider the running time of the algorithm when trying to achieve high probability O(1/n) excess risk bounds. Currently, the algorithm requires n^2 iterations of full-batch which results in O(n^3) gradient-complexity which is significant.

**Main Review:**

The paper uses tools developed by Bousquet et al. [8] heavily to prove their results. It is rather difficult to find a strongly convex function that is Lipschitz continuous and bounded. An easy way to justify these conditions together is to assume that \mathcal{X} is a compact set. The contribution is of good value from a theoretical point of view. As the authors pointed out, the generalized Bernstein assumption might be useful when considering optimization under PL-condition which is widely studied in recent years due to its usefulness for modern ML applications. Overall, the paper makes a good contribution and might serve as a first step to thoroughly explore the new assumption in different settings.

**Time Spent Reviewing:**

6 hours

---

> ### Author Response · Authors · 2021-08-09
> **Answer to the reviewer**
>
> Thank you, it is a good question about the relatively large $O(n^2)$ number of iterations in the non-smooth case. We believe that without the smoothness assumption not much can be improved. In particular, very recently Amir, Koren and Livni (https://arxiv.org/pdf/2102.01117.pdf) considered the problem of convex Lipschitz optimization with a quadratic regularization and showed that in this case, GD requires $\Omega(n^2)$ steps (we thank the reviewer 4(1bRR) for pointing this out). In their Theorem 3.2 the term $\sqrt{\Delta / \lambda}$ appears in the lower bound. Even though they consider a slightly different problem, this number of iterations might not be improvable in our case as well.
>
> Additionally, see the discussions on strongly convex Lipschitz optimization problems in Section 2.

---

> > ### Comment · Reviewer_Bxnk · 2021-08-16
> > **Reply to authors**
> >
> > Thank you for the clarifications.

---

### Author Response · Authors · 2021-08-09
**Response to all reviewers**

We thank all the referees for their appreciation of our work and their useful comments.

---

### Decision · Program_Chairs · 2021-09-27

**Decision:**

Accept (Oral)

**Comment:**

The reviewers are unanimous that this work provides a significant contribution in the area of uniform stability, which should interest the learning theory community as well as the wider NeurIPS community. A key consequence of the authors' results is the positive resolution of a long-standing open question of Shalev-Shwartz et al.'s COLT 2009 paper: this question is about whether ERM can obtain fast rates of convergence for Lipschitz, stochastic strongly convex optimization with high probability, where the dependence on the failure probability $\delta$ is logarithmic. While the authors do heavily draw from previous work, certain technical components like the novel excess risk decomposition are highly original, provide an important technical contribution, and might lead to progress in future works. In addition, the work is very well written and clear. This work would be a welcome contribution to NeurIPS.